# NF-κB Signaling Modulates miR-452-5p and miR-335-5p Expression to Functionally Decrease Epithelial Ovarian Cancer Progression in Tumor-Initiating Cells

**DOI:** 10.3390/ijms24097826

**Published:** 2023-04-25

**Authors:** Rahul D. Kamdar, Brittney S. Harrington, Emma Attar, Soumya Korrapati, Jyoti Shetty, Yongmei Zhao, Bao Tran, Nathan Wong, Carrie D. House, Christina M. Annunziata

**Affiliations:** 1Women’s Malignancies Branch, National Cancer Institute, National Institutes of Health, Bethesda, MD 20892, USA; 2CCR Sequencing Facility, Leidos Biomedical Research, Inc., Frederick National Laboratory for Cancer Research, Frederick, MD 21701, USA; 3Advanced Biomedical Computational Science, Frederick National Laboratory for Cancer Research, Frederick, MD 21701, USA; 4CCR Collaborative Bioinformatics Resource, National Cancer Institute, National Institutes of Health, Bethesda, MD 20814, USA

**Keywords:** epithelial ovarian cancer, NF-κB, *RELA*, *RELB*, miR-452-5p, miR-335-5p

## Abstract

Epithelial ovarian cancer (EOC) remains the fifth leading cause of cancer-related death in women worldwide, partly due to the survival of chemoresistant, stem-like tumor-initiating cells (TICs) that promote disease relapse. We previously described a role for the NF-κB pathway in promoting TIC chemoresistance and survival through NF-κB transcription factors (TFs) RelA and RelB, which regulate genes important for the inflammatory response and those associated with cancer, including microRNAs (miRNAs). We hypothesized that NF-κB signaling differentially regulates miRNA expression through RelA and RelB to support TIC persistence. Inducible shRNA was stably expressed in OV90 cells to knockdown *RELA* or *RELB*; miR-seq analyses identified differentially expressed miRNAs hsa-miR-452-5p and hsa-miR-335-5p in cells grown in TIC versus adherent conditions. We validated the miR-seq findings via qPCR in TIC or adherent conditions with *RELA* or *RELB* knocked-down. We confirmed decreased expression of hsa-miR-452-5p when either *RELA* or *RELB* were depleted and increased expression of hsa-miR-335-5p when *RELA* was depleted. Either inhibiting miR-452-5p or mimicking miR-335-5p functionally decreased the stem-like potential of the TICs. These results highlight a novel role of NF-κB TFs in modulating miRNA expression in EOC cells, thus opening a better understanding toward preventing recurrence of EOC.

## 1. Introduction

Epithelial ovarian cancer (EOC) is a heterogenous disease with a high rate of disease relapse. The stem-like tumor-initiating cell (TIC) population in EOC promotes cancer progression, contributing to its poor prognosis [1,2,3]. We have previously investigated NF-κB signaling pathways for their roles in promoting ovarian cancer TIC function and survival [4]. We showed that the canonical signaling pathway, which is driven by *RELA*, and the non-canonical pathway, which is driven by *RELB*, affect cancer cell proliferation and the cancer stem cell maintenance, respectively [4]. The overlapping and distinct effects of canonical and non-canonical NF-κB signaling suggests the possibility of leveraging its role to find therapeutic targets to treat EOC.

MicroRNAs (miRNAs) comprise a subset of small, non-coding RNAs that regulate gene expression post-transcriptionally. Several miRNAs have been characterized for their association with EOC. miRNAs including miR-18b, miR-590-3p, and miR-21-5p can promote EOC metastasis through various pathways including epithelial-to-mesenchymal transition, sphere formation, and immunomodulation. In contrast, other miRNAs such as miR-101-3p, miR-421, and miR-219-5p may prevent EOC metastasis [5]. Some miRNAs have also been shown to interact with NF-κB activity in the context of EOC, such as miR-130a, which is expressed upon NF-κB signaling to promote EOC proliferation [6]. Other miRNAs, such as miR-9, are thought to suppress EOC invasion by binding to the 3′ untranslated region (UTR) of NF-κB transcripts [7]. miRNAs are an attractive and tangible therapeutic target in helping attenuate EOC growth and metastasis. However, additional work is needed to further identify relationships between miRNA expression, the EOC stem-cell population, and NF-κB signaling within these cells.

In this study, we used miR-seq to identify two candidate miRNAs, hsa-miR-452-5p and hsa-miR-335-5p, which were altered in response to *RELA* or *RELB* silencing in EOC cell lines grown in TIC-enriched spheroid conditions. We characterized the expression of these miRNAs in EOC cell lines that were silenced for either *RELA* or *RELB* and investigated their function by either inhibiting or mimicking the miRNA. We hypothesized that the NF-κB canonical and non-canonical signaling pathways differentially regulate miRNA expression, contributing to TIC survival and overall EOC progression. Therefore, targeting these differentially expressed miRNAs, in conjunction with disrupting NF-κB activity, will hamper EOC TIC survival and reduce disease progression. Here, we experimentally showed that miR-452-5p displayed oncogenic properties in EOC cell lines, was regulated by *RELA* and *RELB*, and, by inhibiting its expression, suppressed the TIC population. Additionally, we found experimentally that miR-335-5p displayed tumor-suppressive properties in EOC cell lines, was regulated by both *RELA* and *RELB*, and, by mimicking its expression, suppressed TIC viability. The work presented in this study demonstrates potential miRNA therapeutic targets downstream of the NF-κB pathway to prevent EOC progression and metastasis.

## 2. Results

### 2.1. miRNAs Are Differentially Expressed in Ovarian Cancer Cells with Altered RELA and RELB Expression

We previously investigated differences in RNA expression between spheroid and adherent growth conditions of OV90 [8] and ACI23 [4] ovarian cancer cells. In this study, miR-seq was performed on OV90 cells grown in adherent culture and in TIC-enriched spheroid culture conditions, with *RELA* and *RELB* knockdown compared to negative control shRNA. We found 443 miRNAs were differentially expressed between adherent and spheroid conditions with *RELA* and *RELB* knockdown. The largest number of differentially expressed miRNAs was in the spheroid conditions, with 138 miRNAs differentially expressed when *RELA* was silenced and 136 when *RELB* was silenced. In adherent conditions, there were 114 differentially expressed miRNAs when *RELB* was silenced, and only 55 when *RELA* was silenced. The differential expression of several miRNAs is plotted based on either *RELA* or *RELB* silencing in adherent or spheroid conditions (Figure 1A–D). Interestingly, few miRNAs were shared between *RELA*- and *RELB*-knockdown samples (Figure 1E), and these miRNAs followed the same direction of expression as up- or down-regulated expression. Of note, seven miRNAs appeared with consistent up- or down-regulated expression across the conditions, including miR-155-5p, which was up-regulated in both adherent and spheroid conditions when *RELA* or *RELB* were knocked down. miRNAs let-7b-5p, miR-100-5p, miR-205-5p, miR-224-5p, miR-552-5p, and miR-675-5p were down-regulated across adherent and spheroid conditions when *RELA* or *RELB* were knocked down. This suggests that a common pathway regulated by NF-κB signaling affects the expression of these miRNAs.

### 2.2. miRNAs 452-5p and 335-5p Expression Is Altered by RELA and RELB Knockdown in EOC Cell Lines

We validated the expression of two miRNAs that were altered by *RELA* or *RELB* knockdown in EOC cell lines. Based on their expression patterns in the miR-seq (Table 1) and the existing literature outlining their potential functionality, we measured the expression of nine miRNAs via qRT-PCR and found that miR-452-5p and miR-335-5p had changes in expression that were consistent with the miR-seq. miR-452-5p has been shown to possess oncogenic characteristics in colorectal cancer, hepatocellular carcinoma, and ovarian carcinoma [9,10,11], while miR-335-5p has been shown to exert tumor suppressive activity in gastric and ovarian cancers [5,12,13]. By using qRT-PCR and probes (Appendix A), we evaluated the expression of miR-452-5p in OV90, the same line used in miR-seq, and two additional ovarian cancer cell lines ACI23 and OVCAR8. The silencing of either *RELA* or *RELB* was also confirmed in these cell lines by using Western blot (Appendix A). The expression of miR-452-5p significantly decreased with *RELA* and *RELB* knockdown, in both adherent and spheroid conditions in OV90 (Figure 2A). Similarly, in ACI23 cells, miR-452-5p expression decreased in *RELA* and *RELB* knockdown in adherent conditions (Figure 2B). In OVCAR8 cells, *RELB* knockdown had a greater effect on suppressing miR-452-5p expression than *RELA* knockdown, and it was more pronounced in spheroid conditions (Figure 2C). We also investigated the expression of miR-335-5p in the same cell lines with *RELA* or *RELB* knockdown in adherent and spheroid conditions and found this to be partially consistent with the miR-seq results. In adherent conditions, we observed an increased expression of miR-335-5p in OV90 with *RELA* knockdown (Figure 2D), which was greater than what was found in the miR-seq results with *RELA* knocked down. No change was observed with *RELB* knockdown. In ACI23 cells, there was a significant increase in miR-335-5p when *RELA* was knocked down in spheroid conditions and decreased expression when *RELB* was knocked down in adherent or spheroid conditions compared to the controls (Figure 2E). In OVCAR8 cells, we saw increased miR-335-5p expression when *RELA* was knocked down in both adherent and spheroid conditions like the OV90s but also significantly increased expression in spheroids with *RELB* knocked down (Figure 2F). Additionally, other candidate miRNAs listed were tested for their expression in OV90 cells initially, and the list was expanded to at least one other EOC cell line (Appendix A). Taken together, these findings suggest that miR-452-5p and miR-335-5p are differentially expressed miRNAs in EOC when either *RELA* or *RELB* are silenced across adherent and/or spheroid conditions.

### 2.3. Inhibition of miR-452-5p and Mimicking miR-335-5p Affects EOC Sphere-Formation Capability

Our previous works have shown that silencing *RELA* or *RELB* decreases the sphere-formation ability of EOC cells [4]. To further assess the functional roles of miR-452-5p and miR-335-5p, we examined the ability of OV90 and OVCAR8 cells to form spheres after silencing either *RELA* or *RELB* and adding either a miR-452-5p inhibitor or miR-335-5p mimic at the maximal and minimal respective effective concentrations (Appendix A). In OV90 spheres with miR-452-5p inhibited, sphere formation was significantly decreased in the shNeg control, *RELA*-silenced, and *RELB*-silenced cells compared to controls, which is consistent with its role as a tumor-promoting miRNA (Figure 3A). Notably, the effect of inhibiting miR-452-5p had a greater relative decrease on sphere formation in the control and *RELA*-silenced cells than in the *RELB*-silenced cells, as the *RELB*-silenced cells had the fewest number of formed spheres (Figure 3A). The knockdown of *RELA* and *RELB* achieved an estimated 70% reduction in *RELA* and *RELB* expression with inducible shRNA silencing (Appendix A), which reduced miR-452-5p expression by approximately 60%. The addition of a miR-452-5p inhibitor has been shown to decrease the relative miR-452-5p expression by 50% (Appendix A). The combined inhibitor and shRNA silencing maximized the miR-452-5p inhibition, for which we observed the affected EOC function. To further validate these findings, we treated OV90 control or *RELA*-silenced cells with a miR-452-5p mimic to rescue the sphere-formation ability. We found that adding the mimic resulted in no significant changes in the number of formed spheres similar to controls in shNeg and *RELA*-silenced cells (Appendix A). In OVCAR8 cells, we observed a significant decrease in sphere-formation ability when the miR-452-5p inhibitor was added to cells with *RELB* silenced; there was a trend toward decreased sphere-formation ability after adding the inhibitor to control or *RELA*-silenced cells, which did not reach statistical significance (Figure 3B). The sphere-formation capability was restored to levels similar to control in OVCAR8 cells with *RELB*-silenced after adding a miR-452-5p mimic (Appendix A). After assessing the miR-452-5p inhibitor on EOC sphere formation, we looked at the effect of mimicking miR-335-5p on EOC sphere formation, based on its hypothesized role as a tumor suppressor. A decreasing trend in the number of spheres was noted more in the OVCAR8 cells than the OV90 cells when miR-335-5p was mimicked compared to the negative control, although it did not reach statistical significance (Figure 3C,D). Taken together, these results support the idea that combining the disruption of NF-κB signaling and modulating miRNA expression helps decrease the sphere-formation ability of EOC TICs.

### 2.4. miR-452-5p and miR-335-5p Differentially Affect EOC Spheroid Viability

We have previously shown that disrupting NF-κB activity decreased the viability of EOC cells grown in spheroid conditions [4]. We sought to characterize the cell viability of EOC cell viability containing *RELA or RELB* knockdown when miR-452-5p and miR-335-5p were inhibited and mimicked, respectively. Consistent with the findings of the RT-qPCR for OV90s, the effect of inhibiting miR-452-5p was pronounced in spheroids with *RELA* and *RELB* knockdown, showing significant reduction in viability compared to negative controls (Figure 4A). The viability of the *RELB*-knockdown spheroids was rescued by restoring miR-452-5p expression by using a mimic; unexpectedly, however, viability was not restored with the miR-452-5p mimic and was further decreased in *RELA*-knockdown spheroids (Figure 4A). Consistent with our previous results [4], we found silencing *RELA* caused a decreased trend in viability compared to *RELB* silencing relative to the shNeg control (Appendix A). In the OVCAR8 cells, spheroid viability was significantly reduced with the miR-452-5p inhibitor in *RELA*-knockdown cells but not in the *RELB*-knockdown spheroids (Figure 4B), in which miR-452-5p levels were reduced the most by qRT-PCR. In assessing the role of miR-335-5p, the viability decreased in OV90 cells with *RELA* silenced (Appendix A). However, mimicking miR-335-5p in OV90 cells significantly increased viability in cells with *RELB* silenced, as well as in shNeg control cells (Appendix A). Upon addition of a miR-335-5p inhibitor, the viability was reduced to levels similar or less than the negative control (NC) condition in both the OV90 *RELB*-silenced and shNeg control cell lines, while the inhibitor rescued the viability effect in OV90 *RELA*-silenced cells (Appendix A). In OVCAR8 spheroids transfected with the miR-335-5p mimic, we observed significantly reduced viability in the *RELB*-knockdown spheroids, which was restored to control when a miR-335-5p inhibitor was used (Appendix A). No significant change in spheroid viability was observed upon adding the miR-335-5p mimic to *RELA*-silenced or shNeg cells; however, the viability was rescued upon the addition of a miR-335-5p inhibitor in *RELA*-silenced cells (Appendix A). These results suggest that the combination of silencing *RELA* and inhibiting miR-452-5p in both OV90s and OVCAR8 cells seems to affect EOC cell viability the greatest, while mimicking miR-335-5p affects viability differently based on the cell line, suggesting a cell-line-specific effect of miRNA expression and viability in spheroids.

### 2.5. miR-452-5p and miR-335-5p Hold No Overall Significant Effect on Anchorage-Independent EOC Colony Formation

Next, we assessed the ability of EOC cells to grow in an anchorage-independent manner by using a soft agar platform. In OV90 cells, the addition of the miR-452-5p inhibitor resulted in a significant decrease in colonies formed when *RELA* was silenced when compared to its control (Appendix A). No significant changes in colony formation were observed in OVCAR8 cells with *RELA* or *RELB* silenced; the only significant decrease in formed colonies was observed from the shNeg control with the addition of an miR-452-5p inhibitor (Appendix A). No significant changes in colony formation were observed in either OV90 or OVCAR8 cells upon addition of the miR-335-5p mimic (Appendix A). Taken together, these findings suggest that when combined with disrupted NF-κB activity, miRNA modulation has a greater effect on some EOC properties such as sphere-formation ability than others such as colony formation.

### 2.6. miR-452-5p Holds a Greater Effect Than miR-335-5p in Targeting TIC Function by Modulating ALDH Activity

In previous works, we have delineated the roles of *RELA* and *RELB* in sustaining the ovarian cancer TIC stem-cell like population while showing that enhanced ALDH activity is a marker for TICs [1,4]. Here, we assessed the roles of miR-452-5p and miR-335-5p on TIC function by measuring ALDH activity in OV90 cells containing either the *RELA* or *RELB* knockdown. For this assay, we did not use OVCAR8 cells, as these cells express low levels of ALDH [1]. In assessing miR-452-5p function on ALDH activity, adding the miR-452-5p inhibitor to *RELA*-silenced cells significantly decreased the percentage of ALDH+ cells compared to cells containing the *RELA* knockdown alone. No significant changes in ALDH activity were observed in *RELB*-silenced cells transfected with the miR-452-5p inhibitor (Figure 5A). Additionally, the addition of a miR-452-5p mimic did not significantly change the percentage of ALDH+ cells back to levels similar to the control in *RELA*-silenced cells (Figure 5B). We next determined the effect of mimicking miR-335-5p on ALDH activity when either *RELA* or *RELB* was silenced. No significant changes in the percentages of ALDH+ cells were observed when *RELA* or *RELB* was silenced and miR-335-5p mimic was added (Appendix A). Taken together, these findings suggest inhibiting miR-452-5p in conjunction with *RELA* silencing most effectively decreases the percentage of ALDH+ cells and serves as a relatively strong target against the broader spheroid population.

### 2.7. SOX7 Is a Target Downstream of miR-452-5p and Its Expression Is Altered in Spheroid EOC Cells

We examined the functional activities of miR-452-5p and miR-335-5p in combination with *RELA* or *RELB* silencing in EOC cells. We observed the consistent effects of miR-452-5p inhibition with RELA silencing on EOC spheroids, whereas the effects of miR-335-5p were variable. This prompted further investigation of the downstream targets of miR-452-5p. Using the miRNA databases microRNA Data Integration Portal (mirDIP) [14,15], we identified and tested the expression of several possible targets of miR-452-5p by using qRT-PCR, including *CDKN1B*, *BEND6*, *BMI1*, *HEYL*, and *SOX7* (Figure 6A). The integrated score from mirDIP represents the confidence scores from their available predictions. Additionally, we found that the miRDB database listed *SOX7* as a target of miR-452-5p with a target score of 81, representing the confidence in the predicted targeting of *SOX7* by miR-452-5p [16,17]. From these results, we selected *SOX7* to further study as a candidate target of miR-452-5p. Furthermore, TargetScanHuman prediction software [18,19] identified the proposed binding site for miR-452-5p on the 3′ UTR of *SOX7* mRNA at nucleotide positions 1845–1852 (Figure 6B). The protein expression of *SOX7* in ovarian cancer was quantified by using immunohistochemistry data from the Human Protein Atlas [20] (proteinatlas.org), in which most cases had a “low” or “not detected” expression of *SOX7* (Figure 6C). In testing the *SOX7* expression in EOC cells, qRT-PCR analysis showed a significant decrease in OVCAR8 cells containing the miR-452-5p mimic (Figure 6D). We also noted a decreasing trend in *SOX7* expression in OV90 cells containing the miR-452-5p mimic as well (Figure 6E). Taken together, these results indicate that miR-452-5p targets and decreases the expression of *SOX7* in EOC, further suggesting a mechanism of *SOX7* regulation through miR-452-5p.

## 3. Discussion

The EOC stem cell population has been associated with a poor prognosis in ovarian cancer, as it promotes chemoresistance, cancer cell invasion, and metastasis. Therefore, studies aimed at targeting these cells are critical for creating lasting therapeutic approaches in EOC. In this study, we examined differential miRNA expression in EOC regulated by NF-κB signaling; we compared the efficacy of disrupting *RELA* or *RELB* signaling and modulating miR-452-5p or miR-335-5p expression on several functional TIC properties. Our results support the proposed roles of miR-452-5p as a tumor-promoting miR and miR-335-5p as a tumor suppressor, each of which are regulated downstream of NF-kB signaling.

From the miR-seq data, we found silencing *RELA* or *RELB* gave rise to a unique set of differentially expressed miRNAs across traditional adherent cell culture and TIC-enriching spheroid culture conditions. We observed an increased number of differentially expressed miRNAs in spheroid conditions than adherent conditions by miR-seq and noted similarities in the differential miRNA expression in our study with miRNAs expressed in other ovarian cancer models. The differential miRNA expression of two-dimensional adherent or three-dimensional spheroid culture conditions in an SKOV3-ipl model were previously reported [21]; their study identified 71 up-regulated and 63 down-regulated miRNAs. In that study, hsa-miR-34a-5p was up-regulated in 3D-sphere-cultured SKOV3-ipl cell lines, and we observed its upregulation in OV90 spheres. The presence of these miRNAs renders them strong candidates to study due to their expression trending in the same direction across different ovarian cancer cell lines grown in spheres. This also suggests a crucial role of miRNA expression in the maintenance and survival of TIC-enriched spheroid cell populations, and further work should examine these spheroid-specific miRNAs.

We were interested in dissecting out the miRNAs regulated by the NF-κB pathways. We identified 274 miRNAs within both *RELA* and *RELB* knockdowns shared between the pathways in the TIC-enriched spheroid condition, but we investigated those that were exclusively altered in response to the silencing of either pathway. From this, miR-452-5p and miR-335-5p were selected for consistent expression changes across EOC cell lines, as well as for largely reflecting the miR-seq trends. The similar miRNA expression trends observed in these cell lines reaffirmed our approach in identifying robust miRNA targets for further characterization.

Based on our current understanding of *RELA*/*RELB* signaling in promoting ovarian cancer proliferation and stem-cell phenotypes [4], the decreased expression of has-miR-452-5p when *RELA* or *RELB* are silenced across OV90, OVCAR8, and ACI23 cells suggests an oncogenic function. Notably, in *RELA*-knockdown cells, adding a miR-452-5 inhibitor caused significant decreases in ovarian cancer cell viability, sphere formation, and percentage of ALDH+ cells. The oncogenic role of miR-452-5p has also been reported in other cancer models including colorectal [9], renal [22], and hepatocellular carcinoma [10]. miR-335-5p has been regarded as a tumor suppressor in cancer models such as gastric [13] and ovarian cancer [23,24]. Interestingly, the upregulation of miR-335-5p increased cisplatin sensitivity within A2780/DDP cell lines [23], abrogating the chemoresistance associated with ovarian cancer stem cells. We observed similar tumor suppressive characteristics of miR-335-5p across several ovarian cancer cell lines silenced for *RELA*/*RELB*. The notable increase in miR-335-5p expression when *RELA* was silenced in OV90s, OVCAR8s, and ACI23 cells supports the idea of either *RELA* or *RELB* contributing to the transcriptional downregulation of miR-335-5p. The increase in miR-335-5p when *RELB* was silenced in OVCAR8 cells, however, suggests that the expression patterns could vary between cell lines and miR-335-5p may be a shared target of both *RELA* and *RELB*. Additionally, the transcriptional regulation by NF-κB proteins on miR-335-5p may occur indirectly through other intermediates rather than a direct binding of the NF-κB proteins onto the miR-335-5p promoter as seen with *RELA* and miR-452-5p [22]. Overall, the observed decrease in miR-452-5p and increase in miR-335-5p expression with *RELA*/*RELB* silencing suggests these targets lie downstream of NF-κB signaling and are regulated by these transcription factors.

Inhibition of miR-452-5p in cell lines silenced for *RELA* functionally attenuated the viability, sphere formation, and ALDH+ population to a greater extent than *RELA* silencing alone. In addition to prior work from our group noting the marked effect of *RELB* signaling on the TIC population [4], we found that combining the miR-452-5p inhibition with *RELA* silencing affected TIC functionality at levels similar to the *RELB*-silenced cells alone, notably in sphere-formation ability and ALDH activity. This is likely due to *RELA* partly constituting the canonical NF-κB signaling pathway, which can be activated by multiple stimuli and could contribute to its dysregulation in a tumor microenvironment [25]. It could also suggest that miR-452-5p has some degree of direct regulation by *RELA* and not *RELB*. One study demonstrated that RelA/p65 could bind to the promoter of miR-452-5p and induce its expression in metastatic renal cell carcinoma, subsequently contributing to renal cell carcinoma invasion and metastasis [22].

Efforts at further identifying and characterizing the target genes regulated by miRNAs such as miR-452-5p will enhance our understanding of the role miRNAs hold in promoting EOC growth and metastasis. Our study found a decreasing trend in *SOX7* expression upon mimicking miR-452-5p. In the greater context of ovarian cancer, *SOX7* has been regarded as a tumor suppressor and a potential negative regulator of the Wnt pathway [26]. Additionally, work conducted by Zheng and colleagues noted that the upregulation of miR-452-5p in patient hepatocellular carcinoma tissues promoted a poor prognosis, partly through increasing chemoresistant properties of hepatospheres. Furthermore, they affirmed the direct targeting of *SOX7* by miR-452-5p by using luciferase reporter assays, while also suggesting that the inhibition of *SOX7* occurs in conjunction with Wnt/β-Catenin pathway activation [27]. While these works suggest a mechanistic link between *SOX7* and Wnt/β-Catenin activity, our future work will build on these by investigating the role miR-452-5p holds in regulating *SOX7* and its downstream targets. This knowledge will support further development of therapeutic agents targeting this signaling axis to improve the outcome of women with EOC.

## 4. Materials and Methods

### 4.1. Reagents

Lipofectamine RNAiMAX transfection reagent; TaqMan^®^ Advanced miRNA Assays (ThermoFisher Scientific, Waltham, MA, USA); miR-452-5p, miR-335-5p, and miR-30e-5p probes; mirVana™ miRNA hsa-miR-452-5p inhibitor (Assay ID MH12509); hsa-miR-452-5p mimic (Assay ID MC12509); hsa-miR-335-5p mimic (Assay ID MC 10063); hsa-miR-335-5p inhibitor (Assay ID MH10063); miRNA inhibitor; and mimic negative controls were from ThermoFisher Scientific (Waltham, MA, USA). RelB antibody (C1E4) was from Cell Signaling Technologies (Danvers, MA, USA). RelA (p65) antibody (10815) was from Santa Cruz Biotechnologies (Dallas, TX, USA), GAPDH antibody (MAB374) was from Millipore Sigma (Burlington, MA, USA).

### 4.2. Cell Lines and Culture Conditions

Ovarian cancer lines OV90 and OVCAR8 were obtained from the American Type Culture Collection (ATCC, Manassas, VA, USA) and were cultured in RPMI 1640 (Thermo Fisher Scientific, Waltham, MA, USA) medium containing 10% (*v*/*v*) fetal calf serum (FCS), penicillin (100 units per mL), and streptomycin (100 units per mL). ACI23 from Dr. John Risinger and Memorial Health University Medical Center, Inc., were maintained in Dulbecco’s Modified Eagle Nutrient Mixture F-12 (DMEM/F12) medium (Gibco, Thermo Fisher Scientific, Waltham, MA, USA) supplemented with 10% FBS, 2 mM L-glutamine and penicillin (100 units per mL), and streptomycin (100 units per mL). shRNA negative, *RELA*, and *RELB* constructs were transfected into OV90 and OVCAR8 cell lines; the induction of *RELA* and *RELB* knockdown occurred after the addition of doxycycline (dox) at a 1:1000 ratio for 72 h. TIC-enriching spheroid culture conditions have been previously described [28]. Briefly, spheroids were generated by maintaining cells in ultra-low attachment (ULA) plates or flasks (Corning, Corning, NY, USA) in defined medium. Experiments involving the TIC-enriched spheroid populations were grown for 3 days in defined medium in ULA plates before treatments were performed. All cultures were maintained at 37 °C in 5% CO_2_.

### 4.3. RNA Extraction and miR-Seq Analysis

Ovarian cancer cell lines were cultured in adherent conditions, and RNA was harvested according to the manufacturer’s instructions (74104, Qiagen, Germantown, MD, USA). miRNA sequencing libraries were prepared by using the QIAseq miRNA Library Kit (Qiagen, Germantown, MD, USA) as follows: A pre-adenylated DNA adapter was ligated to the 3′ ends of miRNAs, and an RNA adapter was ligated to the 5′ end of mature miRNAs. Then, universal cDNA synthesis occurred, in which the reverse transcription (RT) primer contains an integrated unique molecular index (UMI). The RT primer binds to a region of the 3′ adapter and facilitates conversion of the 3′/5′ ligated miRNAs into cDNA while assigning a UMI to every miRNA molecule. During RT, a universal sequence is also added that is recognized by the sample indexing primers during library amplification. After RT, a cleanup of the cDNA is performed by using a streamlined magnetic bead-based method, followed by library amplification. The final purified product was then quantitated by qPCR before cluster generation and sequenced on Illumina NextSeq sequencer with 76 bases single-end reads run set up. The sequencing run was demultiplexed by using Illumina Bcl2fastq v2.20.

The sequencing quality of single-end miR-seq reads was assessed per sample by using FastQC (version 0.11.5) [29] prior to adapter trimming with Cutadapt (version 1.18) [30]. Read quality was subsequently re-evaluated with FastQC, in addition to Kraken (version 1.1) [31] for potential contamination and FastQ Screen (version 0.9.3) [32]. miRDeep2 (version 2.0.0) [33] was then used initially to align sequences to mapped reads available through the miRBase (version 22) [34] reference for the GRCh38 human genome and then used to identify potentially novel microRNA sequences; total reads were subsequently normalized to counts per million. Differential expression in miRNAs was analyzed by using edgeR (version 3.28.1) [35,36]; significantly differentially expressed miRNAs were identified as those with a false discovery rate below 0.01 and an absolute fold change difference greater than 2. miRseq data are deposited in the Gene Expression Omnibus, accession number GSE230678.

### 4.4. qPCR Analysis of miRNAs and Downstream Targets

Total RNA was isolated from cells by using the miRNeasy Mini Kit (Qiagen, Germantown, MD, USA). cDNAs were generated by using the TaqMan Advanced miRNA cDNA Synthesis Kit (ThermoFisher Scientific, Waltham, MA, USA). For downstream targets, cDNA was generated by using the High-Capacity cDNA Reverse Transcription Kit (ThermoFisher Scientific, Waltham, MA, USA). The qPCR was carried out by using a ViiA 7 Real-Time PCR system that used a fast 96-well format (Applied Biosystems, Thermo Fisher Scientific, Waltham, MA, USA). Target probes are listed in Appendix A. The expression levels of target miRNAs were normalized to hsa-miR-30e-5p, an endogenous miRNA control, while the expression levels of *SOX7* (ThermoFisher Scientific Assay ID: Hs00846731_s1) was normalized to GAPDH. Both analyses were conducted by using the 2^−ΔΔCt^ method. Data were collected in triplicate and plotted as the mean with standard deviation (SD). An analysis was performed by using an unpaired *t*-test. * *p* < 0.05, ** *p* < 0.01, *** *p* < 0.001, **** *p* < 0.0001, n.s.—non-significant as compared to Adh or TIC shNeg control.

### 4.5. Transfection of miRNA Inhibitor or Mimic

Adherent cells (2 × 10^5^ cells/well) in 6-well tissue culture plates were transfected with miRNA mimic or were mimic-negative at 1 or 0.1 pM final concentration, or were transfected with inhibitor or were inhibitor-negative at 90 pM final concentration. These concentrations were used after validation by using RT-qPCR analysis (Appendix A). Transfection mixes containing the miRNA, Lipofectamine^®^ RNAi MAX Reagent (Thermo Fisher Scientific, Waltham, MA, USA) in 2 mL Opti-MEM^®^ medium (Invitrogen; Thermo Fisher Scientific, Waltham, MA, USA) were added to individual wells, and cells were incubated overnight. The medium was replaced with RPMI, and cells were allowed to recover for 24 h before subsequent experiments. This was also used for spheroid cells growing in 6-well ultra-low attachment tissue culture plates in stem cell media.

### 4.6. Cell Viability

Cell viability was measured as previously described [1,8] by using CellTiter-Glo (Promega, Madison, WI, USA) according to the manufacturer’s instructions. The analysis was performed by using a two-way ANOVA and Dunnett’s multiple comparisons test. ** *p* < 0.01, **** *p* < 0.0001, n.s.—non-significant as compared to the negative control (NC) within each cell line (shNeg, shRelA, shRelB)

### 4.7. Colony Formation

Colony formation was measured by soft agar assay according to the manufacturer’s instructions (Cell Biolabs, San Diego, CA, USA). Briefly, cells were transfected with negative control or miRNA inhibitor 48 h before plating for colony formation, where a base layer of agar was plated and allowed to solidify before adding the cell-agar layer. The agar layers were topped up with doxycycline-containing media, which was refreshed every 72 h. After 7 days, following the lysis protocol, fluorescence was measured by using a SpectraMax ID3 plate reader (Molecular Devices LLC, San Jose, CA, USA). Data are plotted as mean with SEM and were analyzed by using an unpaired *t*-test. * *p* < 0.05, ** *p* < 0.01, n.s.—non-significant as compared to shNeg, shRelA, or shRelB negative control.

### 4.8. Sphere-Formation Assay

Sphere-formation capability was measured and analyzed by using the methodologies presented in [1] by plating and simultaneously transfecting either OV90 or OVCAR8 cells at 2000 cells/well in a 6-well ULA plate (Corning, Corning, NY, USA). The media was reverted to stem cell media 24 h post-transfection. These cells were re-transfected 72 h after the initial transfection and moved to a 96 well ULA plate with doxycycline-containing media for 72 h. After 7 days after the initial transfection, the sphere-formation capability was measured and analyzed by using the methodologies present in [1]. Analysis was performed by using an unpaired *t*-test. * *p* < 0.05, ** *p* < 0.01, n.s.—non-significant as compared to shNeg, shRelA, or shRelB negative control

### 4.9. ALDH Activity by Flow Cytometry

ALDH activity was determined by using ALDEFLUOR (Stem Cell Technologies, Seattle, WA) according to the manufacturer’s instructions and as described [1,4,8]. Graphs represent data from three independent experiments (*n* = 3) and are plotted to indicate mean with SD. An analysis was performed by using an unpaired *t*-test. ** *p* < 0.01, n.s.—non-significant as compared to negative control.

### 4.10. Western Blot Analysis

Western blot analysis to confirm knockdown of *RELA* or *RELB* in OV90, ACI23, and OVCAR8 cell lines was performed as described in [1], and the results of *RELA* or *RELB* knockdown are shown in Appendix A.

### 4.11. Statistical Analysis

All experimental figures and their statistical analyses were conducted by using GraphPad Prism 8 software (GraphPad, San Diego, CA, USA) using the *t*-test or two-way ANOVA functionality.

## 5. Conclusions

From this study, we identified two miRNAs, miR-452-5p and miR-335-5p, which are differentially expressed in EOC lines containing either *RELA* or *RELB* knockdown in either adherent or spheroid conditions. Based on our experimental validation of the miR-seq by using qRT-PCR and functional assays assessing viability, sphere formation, and ALDH activity, we concluded that miR-452-5p possesses oncogenic properties. Our experimental validation of miR-335-5p noted its increased expression in several EOC cell lines when *RELA* was silenced, indicating its potential tumor suppressive role. Functionally, the use of a miR-335-5p mimic in conjunction with *RELA* and *RELB* silencing significantly affected cell viability. Other assays such as sphere formation showed a decreasing trend in formed spheres when miR-335-5p was mimicked, while no change was noted in ALDH activity or colony formation. The miR-452-5p inhibitor with *RELA* and *RELB* silencing holds the greatest effect in attenuating EOC functionality. To further elucidate targets that are downstream of miRNAs in EOC, we found *SOX7* as a downstream target of miR-452-5p. When EOC cells were transfected with a miR-452-5p mimic, the *SOX7* expression decreased, further affirming the oncogenic properties of miR-452-5p and its ability to regulate *SOX7*, which is regarded as a tumor suppressor in the existing literature. Further characterization of *SOX7* and other gene targets downstream of miR-452-5p will help identify therapeutic approaches to hamper EOC progression.

## Figures and Tables

**Figure 1 ijms-24-07826-f001:**
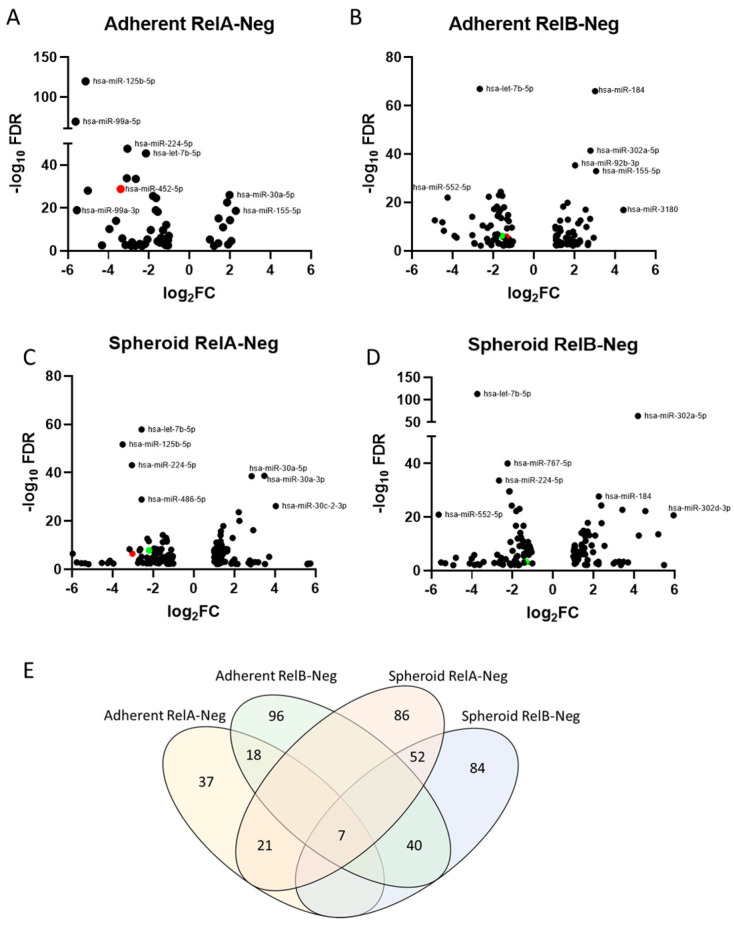
Volcano plot of differentially expressed miRNAs plotted with respect to the log_2_ Fold change (log_2_FC) and −log_10_ False discovery rate (−log_10_FDR) in OV90 cells with: (**A**) *RELA*-knockdown cells grown adherently, (**B**) *RELB*-knockdown cells grown adherently, (**C**) *RELA*-knockdown cells grown in spheroid conditions, (**D**) *RELB*-knockdown cells grown in spheroid conditions. The candidate miRNAs for this study, hsa-miR-452-5p and hsa-miR-335-5p, are identified as red and green circles, respectively. (**E**) Venn diagram of differentially expressed miRNAs in each condition.

**Figure 2 ijms-24-07826-f002:**
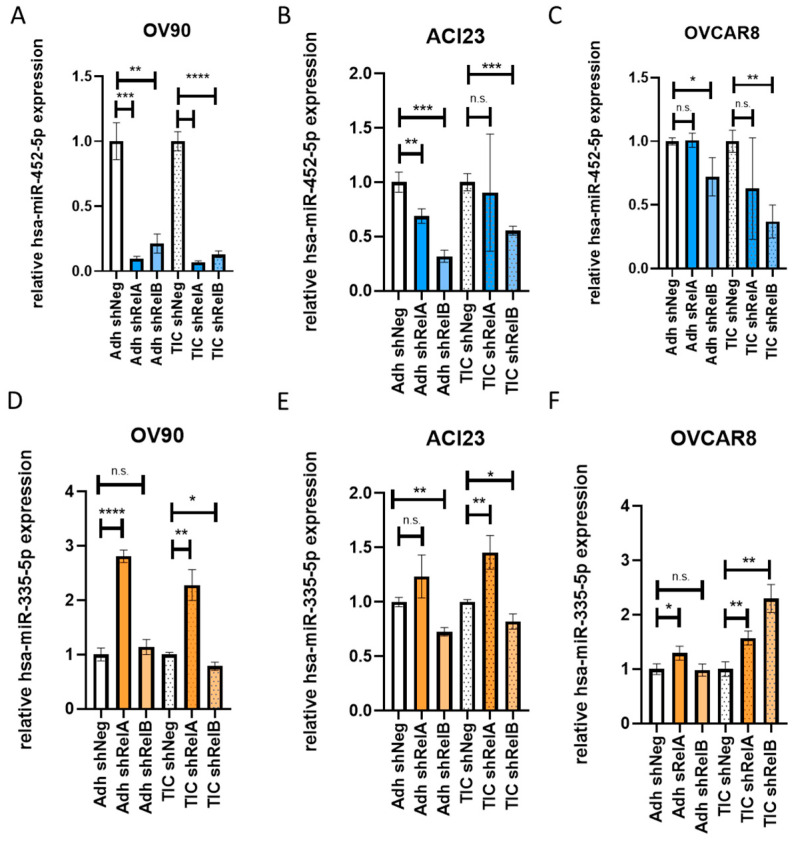
Examining the differential expression of candidate miRNAs in epithelial ovarian cancer cell lines. qRT-PCR analysis showed the expression of hsa-miR-452-5p across adherent (Adh) or tumor-initiating cell (TIC)-enriched spheroid conditions containing either the *RELA* or *RELB* knockdown in OV90 (**A**), ACI23 (**B**), and OVCAR8 (**C**) cell lines. qRT-PCR analysis also showed the expression of hsa-miR-335-5p across Adh or TIC conditions containing either *RELA* or *RELB* knockdown in OV90 (**D**), ACI23 (**E**), or OVCAR8 (**F**) cell lines. Data were collected in triplicate and plotted as the mean with standard deviation (SD). Analysis was performed by using an unpaired *t*-test. * *p* < 0.05, ** *p* < 0.01, *** *p* < 0.001, **** *p* < 0.0001, n.s.—non-significant as compared to Adh or TIC shNeg control.

**Figure 3 ijms-24-07826-f003:**
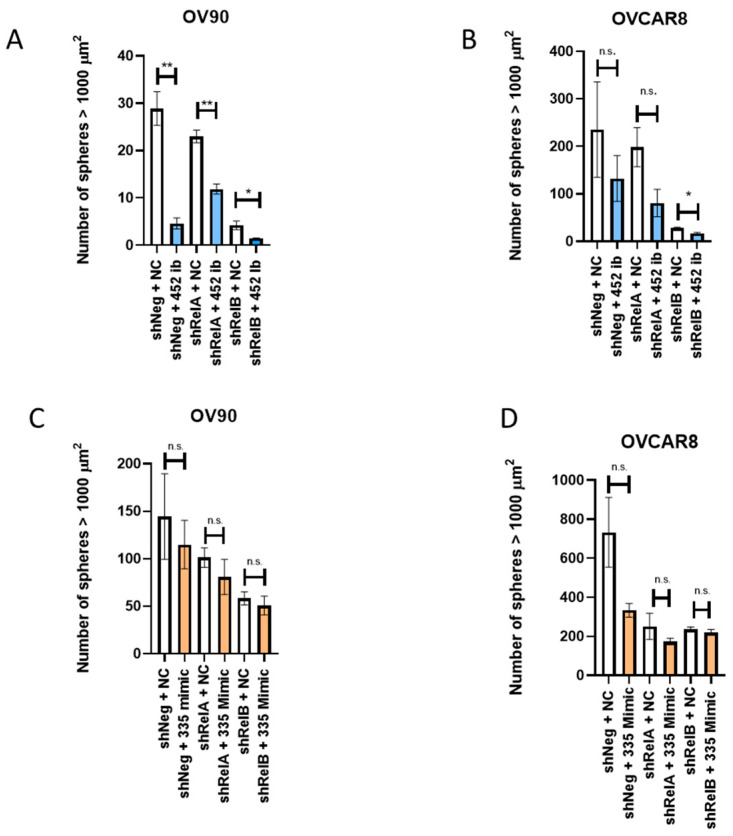
Sphere-formation capability of OV90 and OVCAR8 with either *RELA* or *RELB* silenced. Spheres with an area greater than 1000 square microns were quantified. (**A**) OV90 spheres were transfected with either negative control (NC) or hsa-miR-452-5p inhibitor (452 ib) at 90 pmol. (**B**) OVCAR8 spheres were transfected with either NC or 452 ib. (**C**) Addition of NC or hsa-miR-335-5p mimic (335 mimic) at 1 pmol to OV90 spheres or (**D**) OVCAR8 spheres. Data are plotted as mean with SEM. Analysis was performed by using an unpaired *t*-test. * *p* < 0.05, ** *p* < 0.01, n.s.—non-significant as compared to shNeg, shRelA, or shRelB negative control.

**Figure 4 ijms-24-07826-f004:**
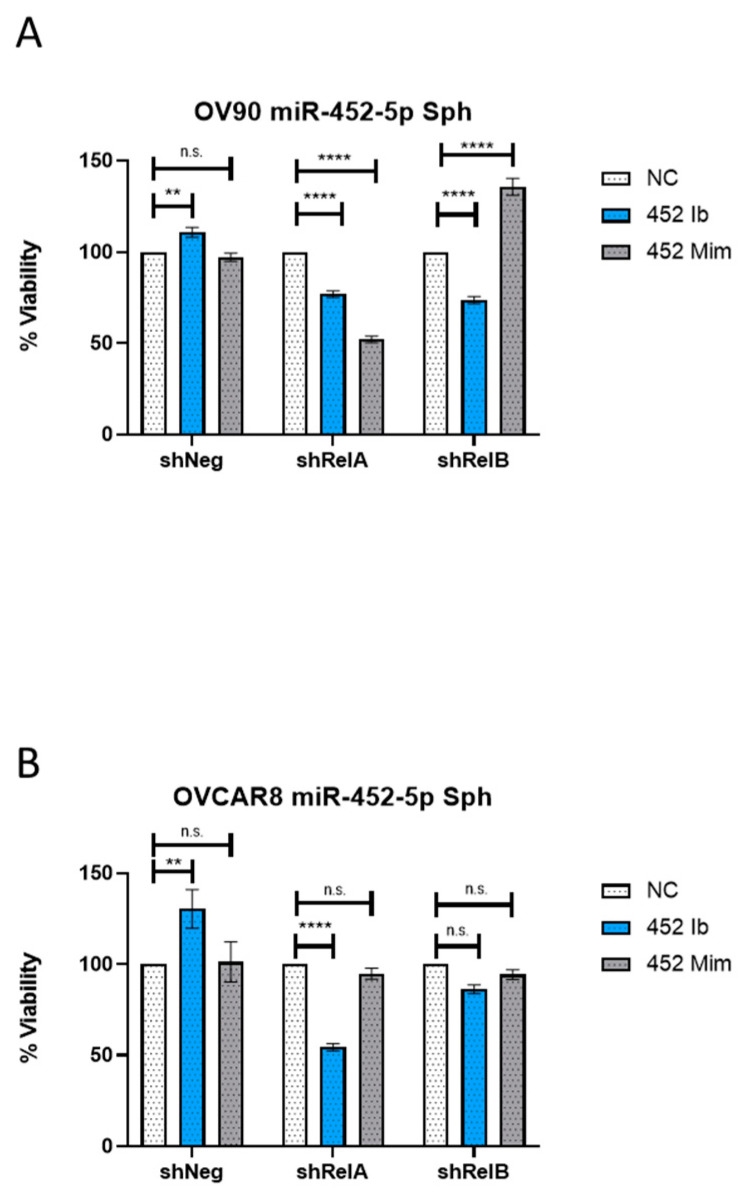
Measured cell viability of EOC cell lines containing silenced *RELA* or *RELB* with modulated miR-452-5p expression. (**A**) Viability of OV90 spheres silenced for either *RELA* or *RELB* and transfected with either 90 pmol of miR-452-5p inhibitor or 0.1 pmol miR-452-5p mimic. (**B**) Viability of OVCAR8 spheres that were silenced for either *RELA* or *RELB* and transfected with either 90 pmol of miR-452-5p inhibitor or 0.1 pmol miR-452-5p mimic. Graphs represent mean with standard error of mean (SEM). Analysis was performed by using a two-way ANOVA and Dunnett’s multiple comparisons test. ** *p* < 0.01, **** *p* < 0.0001, n.s.—non-significant as compared to the negative control (NC) within each cell line (shNeg, shRelA, shRelB).

**Figure 5 ijms-24-07826-f005:**
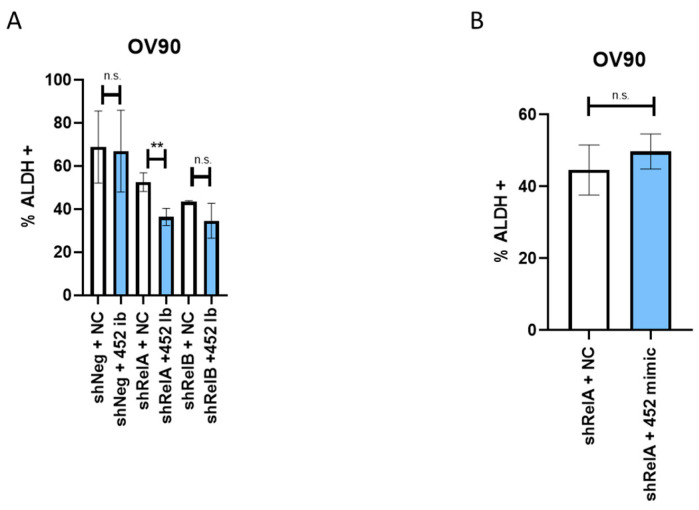
ALDH activity in OV90 EOC cells with silenced *RELA* or *RELB* after miRNA modulation. (**A**) Quantified percentages of ALDH+ cells in *RELA*- or *RELB*-silenced OV90 spheres after adding 90 pmol of either negative control (NC) or hsa-miR-452-5p inhibitor (452 Ib). (**B**) Quantified percentages of ALDH+ OV90 cells with *RELB* silenced and transfected with 0.1 pmol miR-452-5p inhibitor (452 Ib). Graphs represent data from three independent experiments (*n* = 3) and are plotted to indicate mean with SD. Analysis was performed by using an unpaired *t*-test. ** *p* < 0.01, n.s.—non-significant as compared to negative control.

**Figure 6 ijms-24-07826-f006:**
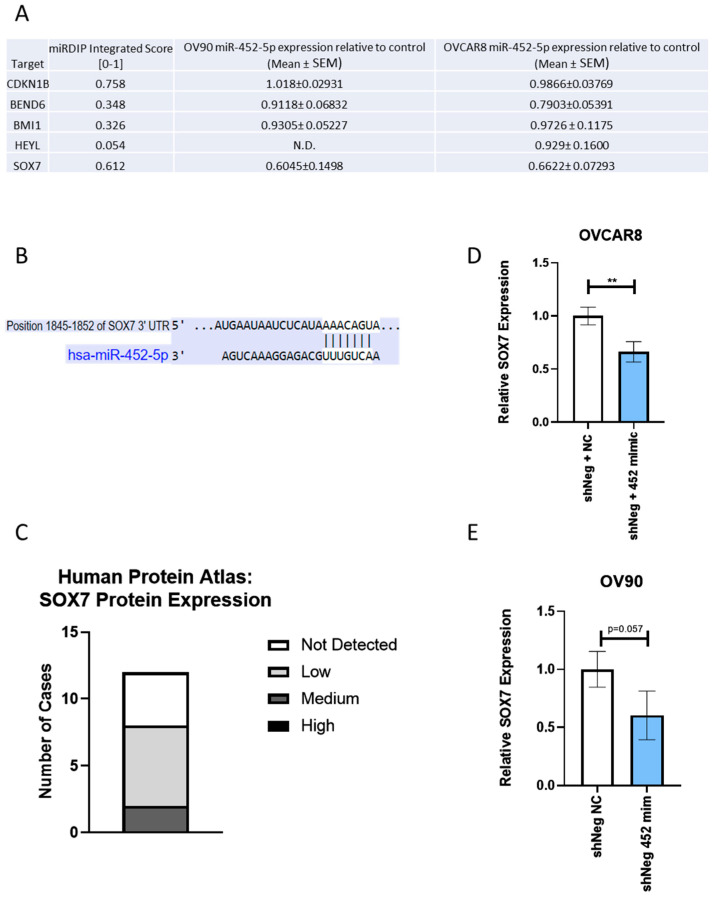
*SOX7* is a putative target downstream of miR-452-5p. (**A**) Table of potential targets downstream of miR-452-5p and their respective integration score from the miRNA Data Integration Portal (mirDIP). qRT-PCR analysis of downstream targets in OV90 and OVCAR8 cells transfected with 0.1 pmol of miR-452-5p mimic also listed as the Mean ± SEM relative to the shNeg + Negative control (NC) transfected cells. N.D. = not detectable. (**B**) Predicted consequential pairing of *SOX7* and miR-452-5p from TargetScanHuman miRNA prediction software. (**C**) Human Protein Atlas immunohistochemistry data displaying *SOX7* expression level by number of cases. (**D**) qRT-PCR analysis of relative *SOX7* expression using OVCAR8 and (**E**) OV90 TIC cDNA transfected with miR-452-5p mimic. Data were collected in triplicate and plotted as mean with SD. ** *p* < 0.01 as compared to shNeg negative control.

**Table 1 ijms-24-07826-t001:** Outline of nine differentially expressed candidate miRNAs from OV90 miR-seq data, along with their respective fold changes (FC) and false discovery rate (FDR) in spheroid conditions when either *RELA* or *RELB* is silenced. The candidate miRNAs that were further studied, hsa-miR-452-5p and hsa-miR-335-5p, are highlighted.

miRNA	Sph RelA-Neg FDR	Sph RelA-Neg FC	Sph RelB-Neg FDR	Sph RelB-Neg FC
hsa-miR-30a-3p	2.10 × 10^−39^	11.177	2.07 × 10^−18^	5.243
hsa-miR-30a-5p	2.58 × 10^−39^	7.229	1.52 × 10^−13^	2.993
hsa-miR-34a-5p	1.59 × 10^−10^	2.256	9.13 × 10^−15^	2.642
**hsa-miR-335-5p**	1.42 × 10^−8^	−4.593	6.07 × 10^−4^	−2.501
hsa-miR-155-5p	7.07 × 10^−17^	7.638	2.03 × 10^−23^	10.655
hsa-miR-200c-3p	4.53 × 10^−9^	2.553	5.00 × 10^−25^	5.205
**hsa-miR-452-5p**	3.79 × 10^−7^	−8.161	6.39 × 10^−2^	−1.956
hsa-miR-105-3p	5.89 × 10^−1^	−1.398	1.46 × 10^−3^	−7.445
hsa-miR-105-5p	3.77 × 10^−2^	1.384	5.44 × 10^−25^	−4.091

## Data Availability

All data are available in the main text or in the Appendix A.

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
