# Peer review of "NF-κB Signaling Modulates miR-452-5p and miR-335-5p Expression to Functionally Decrease Epithelial Ovarian Cancer Progression in Tumor-Initiating Cells"

_ijms, 2023, doi:10.3390/ijms24097826_

Round 1

Reviewer 1 Report

The Authors built up on previous studies demonstrating the role of NF-kB in promoting ovarian cancer TIC functions, and analyzed the effect of RELA and RELB silencing on miRNAs expression in OV90 cells, cultured in adhesion or as spheres. The Authors then selected 2 miRNAs, miR-452-5p and miR-335-5p, that are regulated by RELA or RELB silencing, in adherent and/or spheres, although the expression of these miRNAs in the 3 cell lines analyzed is not always consistent with the Authors’ hypothesis of a NF-kB-regulated expression. Indeed, it is not completely clear why the Authors have selected these 2 miRNAs and not for example some of the 7 miRNAs described at the end of paragraph 2.1, whose expression seemed to be strongly affected by RELA or RELB silencing.

The Authors then analyzed the effect of interfering with miR-452-5p and miR-335-5p on sphere formation, viability and function, in cells silenced for RELA and RELB. However, in these experiments a number of things are not clear or wrongly interpreted:

-in Paragraph 2.3, it is not clear if the Authors assume that the detrimental effect of RELA and RELB silencing on sphere formation is mediated by the two miRNAs. For example, miR-452-5p is downregulated by RELA and RELB silencing, but its inhibitor further potentiates the effect of RELA and RELB silencing on sphere formation. This would suggest that RELA/RELB and the miRNA regulated sphere formation not acting on the same axis. Moreover, the statement at line 165 that the mimic restored a number of spheres similar to controls in shRELA cells is not supported by data in Suppl. Fig. 3A, which actually shows no effect of the mimic on sphere formation. Overall, if the Authors wanted to show that RELA/RELB silencing affected sphere formation regulating miR-452-5p expression, the mimic should have reverted the effect of silencing, while the inhibitor should have exerted no further effect. On the contrary, the Authors are showing an additive effect of RELA/B silencing and miRNA inhibitor, which is interesting, but then it is not clear what is the role of the decreased expression of the miRNA induced by RELA/B silencing.

-in Paragraph 2.6, the statement at line 255 is not supported by data shown in Fig. 5B, which does not show that “miR-452-5p mimic rescued the percentage of ALDH+ cells back to levels similar to the control”, it actually shows no effect of the mimic on percentage of ALDH+ cells.

Finally, the Authors claim that CDKN1B is a target of miR-452-5p and its expression is altered in spheroids silenced for RELA or RELB. These experiments are performed on OV90 cells, in which silencing of both RELA and RELB strongly downregulated miR-452-5p expression. With this in mind, it is not clear why in shRELA cells CDKN1B was downregulated only in presence of miR-452-5p mimic. Moreover, if CDKN1B was a direct target of miR-452-5p, why the miR-452-5p mimic was not able to downregulate its expression in control cells?

Minor: some refs in the Introduction are not cited appropriately, for example:

-refs 4 and 5 cited at lines 42 and 44 are not about NF-kB. Refs 7 would be more appropriate

-again ref 4 at lines 54 and 56 should probably be replaced by PMID: 28800130 and PMID: 19702828, respectively.

Reviewer 2 Report

1. Title and focus of the study on nfkb and so much work of the miRNAs presented either these two miRNAs be listed in the title or made a catchy title representing the complete study.

2, miRNAs focus was good in relation to nfkb but focused and necessary details are required.

3. Discussion needs focus and relevant 

4. readers get confused that too much work has been done and the conclusion should be sound according to the detailed work done

5. Too much good study and very well organized just need to the point and relevant information to be put in

Reviewer 3 Report

The authors found that NF-kB signaling can regulate the expression level of hsa-miR-452-5p and hsa-miR-335-5p to contribute the stemness of TICs.

Major issues:

You can expand your validation experiment to more cell lines such as Kuramochi.

You can also test the migration and invasion capabilities of effects of those miRNAs.

In NF-kB signaling pathway, we have other members. It will be better if you can also explore the associations of them and miRNAs identified in the paper.

Round 2

Reviewer 1 Report

I am satisfied with the Authors' answers to my questions.

Reviewer 3 Report

It can be accepted in the present form.